# Solute Transport and Transformation in an Intermittent, Headwater Mountain Stream with Diurnal Discharge Fluctuations

**Adam S. Ward [1],\*** , **Marie J. Kurz [2,3]**, **Noah M. Schmadel [4]**, **Julia L.A. Knapp [5,6]**,
**Phillip J. Blaen [7,8]**, **Ciaran J. Harman [9]**, **Jennifer D. Drummond [7]**, **David M. Hannah [7]**,
**Stefan Krause [7,10]**, **Angang Li [11]**, **Eugenia Marti [12]**, **Alexander Milner [7]**, **Melinda Miller [1]**,
**Kerry Neil [1]**, **Stephen Plont [13,14]**, **Aaron I. Packman [11]**, **Nathan I. Wisnoski [15]**,
**Steven M. Wondzell [16] and Jay P. Zarnetske [14]**

[1]   O'Neill School of Public and Environmental Affairs, Indiana University, Bloomington, IN 47401, USA;
     mgerhart@indiana.edu (M.M.); kerryneil17@gmail.com (K.N.)
[2]   The Academy of Natural Sciences of Drexel University, Philadelphia, PA 19104, USA; mk3483@drexel.edu
[3]   Department of Hydrogeology, Helmholtz Centre for Environmental Research-UFZ, 04318 Leipzig, Germany
[4]   U.S. Geological Survey Earth System Processes Division, Reston, VA, 20192, USA; nschmadel@usgs.gov
[5]   Department of Environmental Systems Science, ETH Zurich, 8092 Zurich, Switzerland;
     julia.knapp@usys.ethz.ch
[6]   Center for Applied Geoscience, University of Tübingen, 72074 Tübingen, Germany
[7]   School of Geography, Earth & Environmental Sciences, University of Birmingham, Edgbaston,
     Birmingham B15 2TT, UK; pblaen@gmail.com (P.J.B.); J.Drummond@bham.ac.uk (J.D.D.);
     D.M.HANNAH@bham.ac.uk (D.M.H.); S.Krause@bham.ac.uk (S.K.); a.m.milner@bham.ac.uk (A.M.)
[8]   Yorkshire Water, Bradford BD6 2SZ, UK
[9]   Department of Environmental Health and Engineering, Johns Hopkins University, Baltimore, MD 21218,
     USA; charman1@jhu.edu
[10]  University Claude Bernard Lyon 1, LEHNA - Laboratory of Ecology of Natural and Man-Impacted
     Hydrosystems, 69622 Lyon, France
[11]  Department of Civil and Environmental Engineering, Northwestern University, Evanston, IL 60208, USA;
     angang-li@u.northwestern.edu (A.L.); a-packman@northwestern.edu (A.I.P.)
[12]  Integrative Freshwater Ecology Group, Center for Advanced Studies of Blanes (CEAB-CSIC), 17300 Blanes,
     Spain; eugenia@ceab.csic.es
[13]  Department of Biological Sciences, Virginia Polytechnic Institute and State University, Blacksburg, VA 24061,
     USA; plontste@vt.edu
[14]  Department of Earth and Environmental Sciences, Michigan State University, East Lansing, MI 48824, USA;
     jpz@msu.edu
[15]  Department of Biology, Indiana University, Bloomington, IN 47405, USA; wisnoski@indiana.edu
[16]  US Forest Service, Pacific Northwest Research Station, 3200 SW Jefferson Way, Corvallis, OR 97331, USA;
     steve.wondzell@usda.gov
\*    Correspondence: adamward@indiana.edu

**Abstract:** Time-variable discharge is known to control both transport and transformation of solutes in the river corridor. Still, few studies consider the interactions of transport and transformation together. Here, we consider how diurnal discharge fluctuations in an intermittent, headwater stream control reach-scale solute transport and transformation as measured with conservative and reactive tracers during a period of no precipitation. One common conceptual model is that extended contact times with hyporheic zones during low discharge conditions allows for increased transformation of reactive solutes. Instead, we found tracer timescales within the reach were related to discharge, described by a single discharge-variable StorAge Selection function. We found that Resazurin to Resorufin (Raz-to-Rru) transformation is static in time, and apparent differences in reactive tracer were due to interactions with different ages of storage, not with time-variable reactivity. Overall we found

reactivity was highest in youngest storage locations, with minimal Raz-to-Rru conversion in waters older than about 20 h of storage in our study reach. Therefore, not all storage in the study reach has the same potential biogeochemical function and increasing residence time of solute storage does not necessarily increase reaction potential of that solute, contrary to prevailing expectations.

**Keywords:** unsteady-state discharge; solute transport; intermittent stream; diurnal discharge fluctuations; reactive tracers; headwaters; river corridor; hyporheic; resazurin

## 1. Introduction

Discharge dynamics are known to control both transport and chemical transformations in the river corridor. Discharge is often considered a master variable in river corridors because it directly controls residence times of river water, and thus, many chemical reaction processes [1,2]. However, there are a wide range of interactions between transport and transformation, including changes in the relative importance of downstream transport and hydrologic exchange flows [3,4], changes in timescales of contact with reactive storage zones [5,6], and changes in reactivity as streams wet and dry [7]. Despite the recognized role of discharge as a primary control on both transport and transformation processes individually, few studies have considered the joint responses of transport and transformation to dynamic hydrologic forcing. Thus, our overarching goal in this study is to explain how daily fluctuations in discharge control both transport and transformation of reactive tracers in an intermittent stream reach.

Discharge-related hydrological forcing influences river corridor exchange fluxes between river water and reactive sediments and related subsurface transit time distributions, such as in response to storm events [8–12], tides [13–16], baseflow recession [5,17–22], diurnal fluctuations due to evapotranspiration in the catchment and river corridor [22–24], glacial melt or snowmelt [25–28], dam releases [29–32], or wastewater treatment plant operations [33]. Even in intermittent streams where all discharge can become subsurface, discharge dynamics may lead to variation in travel times through the study reach. Dynamics in sediment porewater flow arise due to variation in hydraulic gradients that can cause compression or expansion of reactive storage zones as discharge rises and falls [34–45]. Changes in discharge also affect mixing with in-stream storage [46–49], initiate bank storage [50,51], and possibly change the geometry of subsurface flow paths themselves [52,53]. Overall, we expect the relative role of storage to be maximized during periods of lowest discharge, resulting in extended transit times through the study reach. For example, during low flow conditions, we expect the contact between stream water and the streambed to be maximized due to: (1) the largest bed area per unit discharge [54–56]; (2) exchange flux representing the largest fraction of streamflow [4,36,57]; and (3) less light attenuation with depth in the water column, as would be expected during higher flows [58–60]. Intermittent flow ensures that neither water nor solutes are able to bypass the reactive streambed and hyporheic zone, eliminating the potential shunting of unreacted solutes through the study reach [61].

Reaction rates in the river corridor vary directly in response to dynamics in discharge because they scale with the ratio of river water volume to the biogeochemically active river bed [62]. Stream hydraulics, therefore, may control which microbial or algal communities thrive and thus contribute to high reaction rates [63–65]. High flows may scour streambeds and catalyze either a net reduction in reach-scale reactivity as the microbial community recolonizes or stimulate productivity due to delivery of limiting reagents [66–69]. Reaction rates are known to vary in response to wetting and drying of streambeds [7], where oxidizing and reducing conditions are varied in response to water levels in intermittent streams. The major mechanisms of disturbance-recovery (e.g., scour, desiccation) that affect metabolic activity after large perturbations may not be dominant during regular wetting and drying of diurnal fluctuations in discharge, because diurnal wetting and drying is more frequent and

less extreme than large disturbances. However, the changes in contact times with reactive surface area and fluctuations in solute availability may still drive diurnal changes in metabolic activity.

Solute tracers are commonly used to measure reach-scale transport and transformation processes in river corridors, but usually during low flow conditions with steady discharge. One tool that has proven effective in assessment of combined transport and transformation is the Resazurin–Resorufin (hereafter Raz–Rru, respectively) tracer system, in which Raz undergoes an irreversible transformation to Rru in the presence of metabolically active microbial communities [70–73]. The Raz–Rru system has broadly been used to identify the portion of transient storage that is metabolically active, and has been used to assess reactivity in biofilms [74], the benthic zone [75,76], vegetation beds [77], stream- and lakebed sediments [78–81], and whole stream reaches [82–85]. Interpretation of Raz–Rru information commonly relies upon inverse modeling that enforces a conceptual model with binary divisions of the system into zones such as channel or storage (based on the importance of advection and relative timescales of storage) or metabolically active or inactive storage (based on transformation rates) [83,86].

One challenge with modeling solute tracer transport and transformation is parameter uncertainty and equifinality [18,86–90]. Further, in their standard form, these models have not been formulated to address time-variable transport or transformation, and do not account for the possibility that the system is more complex than the binary differentiation of stream and storage zone. Moreover, common model formulations are not appropriate for interpreting experimental data in intermittent streams, as they require the presence of a continuously flowing stream [91–93], though exceptions do exist [23]. To overcome these limitations, new approaches, such as the StorAge Selection (SAS) model, take transport as a continuum and can simulate continuously variable discharge [94–96].

The overall objective of this study is to assess how reach-scale transport and transformation of reactive tracers vary in response to diurnal discharge fluctuations in an intermittent, headwater stream. We expect that transit times will be longest and reach-scale transformation highest during periods of low discharge because water and solute tracers will experience extended contact with highly reactive streambeds and hyporheic zones. Conversely, during higher-discharge periods, we expect water and solutes to have less contact with hyporheic zones, because previously stored water will be upwelling to the stream, compressing hyporheic flowpaths and forcing tracer to remain in and near the stream channel in shortened flow paths. To test these hypotheses, we conducted a series of four solute tracer experiments, including both conservative and reactive solute tracers (Uranine and Resazurin, respectively) during a period when evapotranspiration was causing diurnal fluctuations in discharge. We also measured in-stream anion and cation concentrations during a 34-h period to assess changes in transport and transformation that may be apparent in fluctuations of the ambient water chemistry.

## 2. Methods

### 2.1. Site Description

Experiments were conducted in Watershed 01 (WS01) in the H.J. Andrews Experimental Forest (Western Cascade Mountains, Oregon, USA) during July and August 2016. The study stream is steep and characterized by a pool–step morphology. The valley is bedrock-constrained both vertically and laterally and overlain by colluvium deposited from hillslope failures. The catchment and study reach have been investigated extensively and well characterized in past field and modeling studies [5,23,53,84,97–102].

For this study, we established an experimental reach within WS01 of about 60 m in length. Pressure transducers (U20L-04 model, Onset) measured stream stage at three locations within the study reach at 15-min intervals (see additional details in related publications [5,53]). The study reach was selected to include a reach that would be spatially intermittent during the entire study period, ensuring that all stream water and solutes would turnover through the hyporheic zone. The site includes a large gravel wedge and step through which 100% of streamflow is transported through the subsurface. Vegetation in the basin is primarily red alder in the valley bottom, with stands of Douglas fir of

varied ages and shrubs on the hillslopes [24]. Stream discharge varied due to evapotranspiration in the basin [5,22–24,95,103–108], but contiguous surface flow along the reach was never observed (i.e., the reach was spatially intermittent for the entire study period). We monitored the maximum and minimum extent of surface water during the study, flagging the most expansive and contracted conditions within the study reach and mapping their positions with a topographic survey (Figure 1).

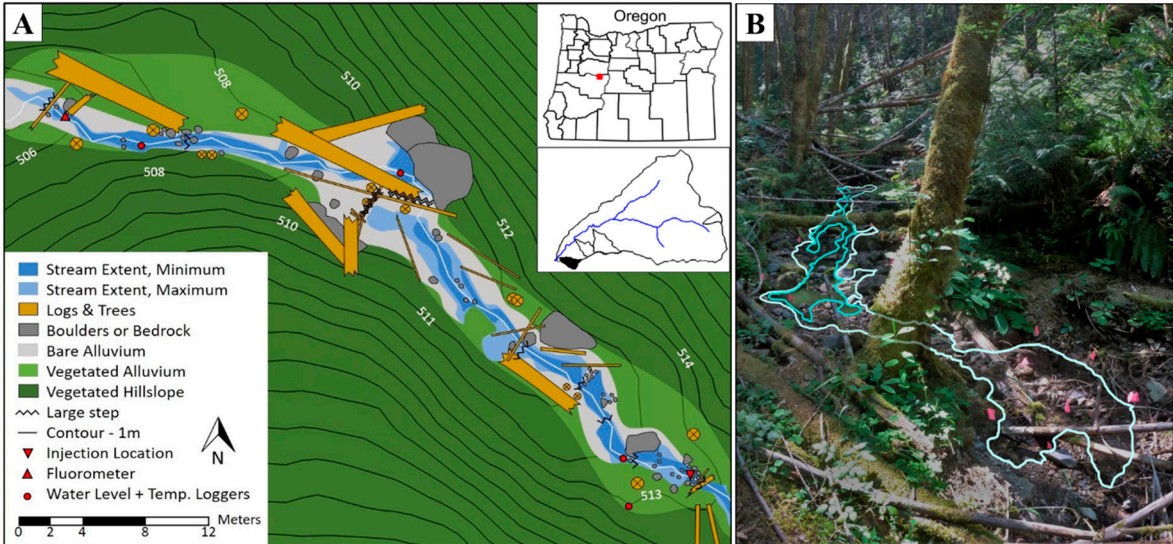

**Figure 1.** (**A**) Plan-view map of the study reach, including surface water extent at the diurnal minimum and maximum, channel morphology, reach topography, and logger locations. (**B**) Image looking upstream at the large step in the middle of the reach. Annotation added to show the wetted perimeter at the diurnal maximum (light blue) and minimum (darker blue) extent.

## 2.2. Stream Solute Tracer Experiments

To measure reach-scale transport and transformation processes, we conducted a series of four replicate solute tracer (both conservative and reactive) injections between 29 July and 7 Aug-2016, a baseflow period free of precipitation. During the study period, diurnal peak and minimum discharges occurred at approximately 13:00 and 04:00, respectively (Figure 2G–I). These instantaneous injections were timed to start at minimum, rising, peak, and falling discharge conditions, with each tracer test measured for more than 24 h, and thus each tracer test underwent the full range of diurnal discharge fluctuations. For each injection, we released known masses of NaCl, Resazurin (Raz), and Uranine (Ura) at a location about 550 m upstream from the weir in WS01 (Figure 1A; Table 1). We monitored in-stream concentrations of Uranine, Resazurin, and Resorufin (Rru; daughter product of Raz) at 10-sec frequency at a location about 55 m downstream of the injection point (GGUN-FL30 Fluorometer, Albilla, Inc.). Sensors for Ura, Raz, and Rru were calibrated using a laboratory fluorometer and standards made with stream water from the site. Grab samples were collected regularly throughout the injections and run on the laboratory fluorometer to correct for drift.

We measured in-stream fluid conductivity about 5 m downstream of the injection location and converted measured specific conductance to concentration of NaCl using a calibration curve developed by adding known masses of NaCl to water from the study site. These observations were used to calculate stream discharge via dilution gauging for the minimum, rising, and peak discharge conditions prevailing during the different tracer injections (logger failure on falling limb injection). The stream channel above and within the 5-m dilution gaging reach is primarily bedrock, giving us confidence that these measurements captured the majority of the water flowing through the valley. We used stage measurements from an in-stream water level logger co-located with the fluid conductivity monitoring location to construct a linear stage–discharge curve from these three discharge values.

Using the regression and the 15 min stage record, we estimated continuous discharge throughout the study period.

We interpreted Ura time series to describe conservative transport through the study reach. While Ura is known to photodecay, the study reach is well-shaded and has a heavy tree canopy. As a proxy for reactive transport and metabolic activity through our reach, we assessed Raz and Rru time series. Raz-to-Rru transformation is a single reactive pathway commonly associated with aerobic respiration and is a reliable proxy for respiration in our study system [83,84,109].

**Table 1.** Summary of tracer injections.

| Injection Code | Date and Time | NaCl (g) | Ura (g) | Raz (g) | Q at Release (L s$^{-1}$) |
|:---:|:---:|:---:|:---:|:---:|:---:|
| Falling | 17:15 29-July-2016 | 899.8 | 2.504 | 8.003 | 0.70 |
| Rising | 05:02 1-August-2016 | 599.0 | 2.504 | 8.005 | 0.59 |
| Peak | 10:39 3-August-2016 | 601.2 | 2.495 | 8.002 | 0.76 |
| Minimum | 23:52 5-August-2016 | 698.8 | 2.501 | 7.999 | 0.24 |

To characterize differences between the observed Ura breakthrough curves, we calculated a series of summary metrics to describe the transport. Notably, we used mass flux rather than concentration as the basis for these calculations to account for the time-variable discharge in the system. First, we calculated the time at which 99% of the recovered tracer exited the reach ($t_{99}$) and the travel time for the peak through the study reach ($t_{peak}$) based on the observed solute tracer timeseries. Next, observed concentrations ($C(t)$) and discharges ($Q(t)$) were converted to normalized mass flux ($m(t)$) as:

$$m(t) = \frac{Q(t)C(t)}{\int_{t=0}^{t=t_{99}} Q(t)C(t)dt} \tag{1}$$

The first temporal moment ($M_1$) is the median travel time through the study reach, calculated as:

$$M_1 = \int_{t=0}^{t=t_{99}} tm(t)dt \tag{2}$$

Next, 2nd and 3rd order central temporal moments ($\mu_2$ and $\mu_3$) were calculated as:

$$\mu_n = \int_{t=0}^{t=t_{99}} (t - M_1)^n m(t)dt \tag{3}$$

where $n$ is the moment order. These moments can be combined to describe the symmetrical spreading normalized to travel time (coefficient of variation, $CV$) and asymmetrical late-time tailing (skewness, $\gamma$) as:

$$CV = \frac{\mu_2^{1/2}}{M_1} \tag{4}$$

and

$$\gamma = \frac{\mu_3}{\mu_2^{3/2}} \tag{5}$$

Additionally, we calculated holdback ($H$), which places the tracer in a range of plug flow ($H = 0$) to only dispersive transport ($H = 1$) [109], where higher values of $H$ indicate increasing importance of non-advective transport processes:

$$H = \frac{1}{M_{1,norm}} \int_{t=0}^{M_1} F(t)dt \tag{6}$$

$$F(t) = \int_{\tau=0}^{t} m(\tau)d\tau \tag{7}$$

where $F$ is a dummy variable for integration. Finally, we calculated apparent dispersion ($D_{app}$) as:

$$D_{app} = \frac{\mu_2 L^2}{2M_1} \tag{8}$$

where $L$ is the length of the study segment.

### 2.3. StorAge Selection Modeling

We used the StorAge Selection (SAS) approach to interpret conservative solute transport through the study reach [94–96]. Briefly, the SAS approach considers outflow as a combination of waters sampled from different ages within the total storage volume between the injection and observation locations. The approach does not require an arbitrary division of surface from subsurface water, nor does it force observations to be fit by various transport mechanisms (e.g., advection, dispersion). Furthermore, this approach allows us to examine the potential importance of down-valley underflow (i.e., down-valley flow in the subsurface [23,110–112]) by incorporating this as an additional flux through the stream reach whose rate we hope to infer from the data. Importantly, the SAS approach is time-variable, whereas the summary metrics we calculated above (e.g., temporal moments) are integrative through time. Here, we closely follow the approach of Harman et al. [95] in describing the SAS function for the channel ($\Omega_C$) as a shifted gamma distribution with offset $S_{min}$, shape parameter $\alpha$, and scale parameter $\beta$ (which is related to the mean, $\mu$, as $\beta = \mu/\alpha$):

$$\Omega_C(S_T, t) = \frac{\gamma\left(\alpha, \frac{S_T - S_{min}}{\beta}\right)}{\Gamma(\alpha)} \quad (S_T > S_{min}) \tag{9}$$

where $\Gamma(\alpha)$ and $\gamma(\alpha, x)$ are the complete and incomplete gamma functions. In keeping with the conceptual model of Ward et al. [23], we represented the down-valley underflow with a separate SAS function ($\Omega_U$), assumed to be an exponential distribution with scale parameter ($S_U$):

$$\Omega_U(S_T, t) = \frac{1}{S_U}exp\left(-\frac{S_T}{S_U}\right) \tag{10}$$

The SAS parameters and underflow discharge ($Q_U$) are each linked to the observed channel discharge by the relationships:

$$S_{min}(t) = S_{minref}exp\left(k_{S_{min}}(Q - Q_{ref})\right) \tag{11}$$

$$\alpha(t) = \alpha_{ref}exp\left(k_\alpha(Q - Q_{ref})\right) \tag{12}$$

$$\mu(t) = \mu_{ref}exp\left(k_\mu(Q - Q_{ref})\right) \tag{13}$$

$$S_U(t) = S_{Uref}exp\left(k_{SU}(Q - Q_{ref})\right) \tag{14}$$

$$Q_U(t) = Q_{Uref}exp\left(k_{QU}(Q - Q_{ref})\right) \tag{15}$$

where the reference discharge, $Q_{ref}$, is 0.55 L s$^{-1}$, (the approximate mean discharge during the period) at which the parameters take on values $S_{min,ref}$, $\alpha_{ref}$, $\mu_{ref}$, $S_{Uref}$, and $Q_{Uref}$. The sensitivity parameters $k_{Smin}$, $\alpha_{ref}$, and $\mu_{ref}$ determine the direction and strength of the relationship between the parameters and discharge and can be positive or negative.

Parameters were inferred by Bayesian parameter estimation using DREAM [113,114]. Model likelihoods were estimated by comparing the log likelihood of each parameter set to observed conservative transport of Ura through the study reach, assuming a Gaussian error ($\sigma_{error}$). This error was inferred alongside the model parameters. DREAM was run with 12 chains for 100,000 generations each. After 10,000 generations, convergence was observed and prior generations were discarded. For

the majority of the analysis below, the maximum likelihood parameter set was used. We report results as the maximum likelihood parameter set, and the 10th through 90th percentile range of the parameters in the retained generations was used to evaluate parameter identifiability.

*2.4. In-Stream Biogeochemistry*

To test for diurnal fluctuations in solute concentrations that might be associated with fluctuations in discharge, we collected samples from the stream channel just above the tracer injection site every 2 h for 34 h, from 10:00, 3 Aug 2016 to 14:00, 4 Aug 2016, contemporaneously with the third solute tracer injection. All sample collection, preservation, and analyses follow the methods described by Ward et al. [114]. Briefly, aliquots were filtered on-site for nutrient and major ion analysis using a 0.45 μm PES filter and for carbon analysis using a 0.2 μm cellulose acetate filter into 60 mL HDPE bottles. All bottles and syringes were triple rinsed with sample water. All samples were chilled immediately after collection, and nutrient/major ion samples were frozen within 24 h of collection until analysis. Samples were analyzed for anions ($Cl^-$, $SO_4^-$; Dionex Ion Chromatograph, University of Birmingham), nutrients ($NO_3^-$, $NH_3$, $PO_4^{3-}$; Skalar Nutrient Analyzer, University of Birmingham), and dissolved organic carbon (DOC; Shimadzu TOC-Analyser, Michigan State University). Finally, we used Mann–Kendall tests to evaluate the likelihood of monotonically increasing or decreasing trends relating discharge to in-stream concentration for each analyte, interpreting $p > 0.10$ as indicating that a monotonic trend between discharge and concentration is unlikely.

## 3. Results and Discussion

*3.1. Conservative Solute Tracer Transport Fluctuates Consistently with Unsteady Discharge*

Observations of conservative tracer (Ura) isolate the role of time-variable discharge on residence times within the study reach. In-stream time series of concentrations (Figure 2A) and mass flux (Figure 2D) are visually consistent between injections, indicating that diurnal discharge fluctuations are controlling the timing of conservative mass transport and storage. In all four injections (falling, rising, peak, minimum), mass flux is reduced at the daily minimum discharge conditions, and rebounds rapidly when discharge and surface water extent increase. For the injections conducted on the rising and peak discharge conditions, the highest concentrations are advected through the study reach so rapidly that they do not experience minimum discharge conditions (i.e., peaks prior to minimum discharge; Figure 2A). As a result, the highest concentration water was preferentially exported from the study reach under higher discharges rather than being stored. Consistent with this preferential export, the late-time tail is no longer visible as the second rising limb of discharge occurs (X = 60 h since 00:00 on day of injection; Figure 2A). In contrast, the peak concentration water from the tracer test remained in the study reach during the minimum discharge conditions (Figure 2A). Consequently, peak concentration water was stored in the reach rather than rapidly exported during minimum discharge conditions.

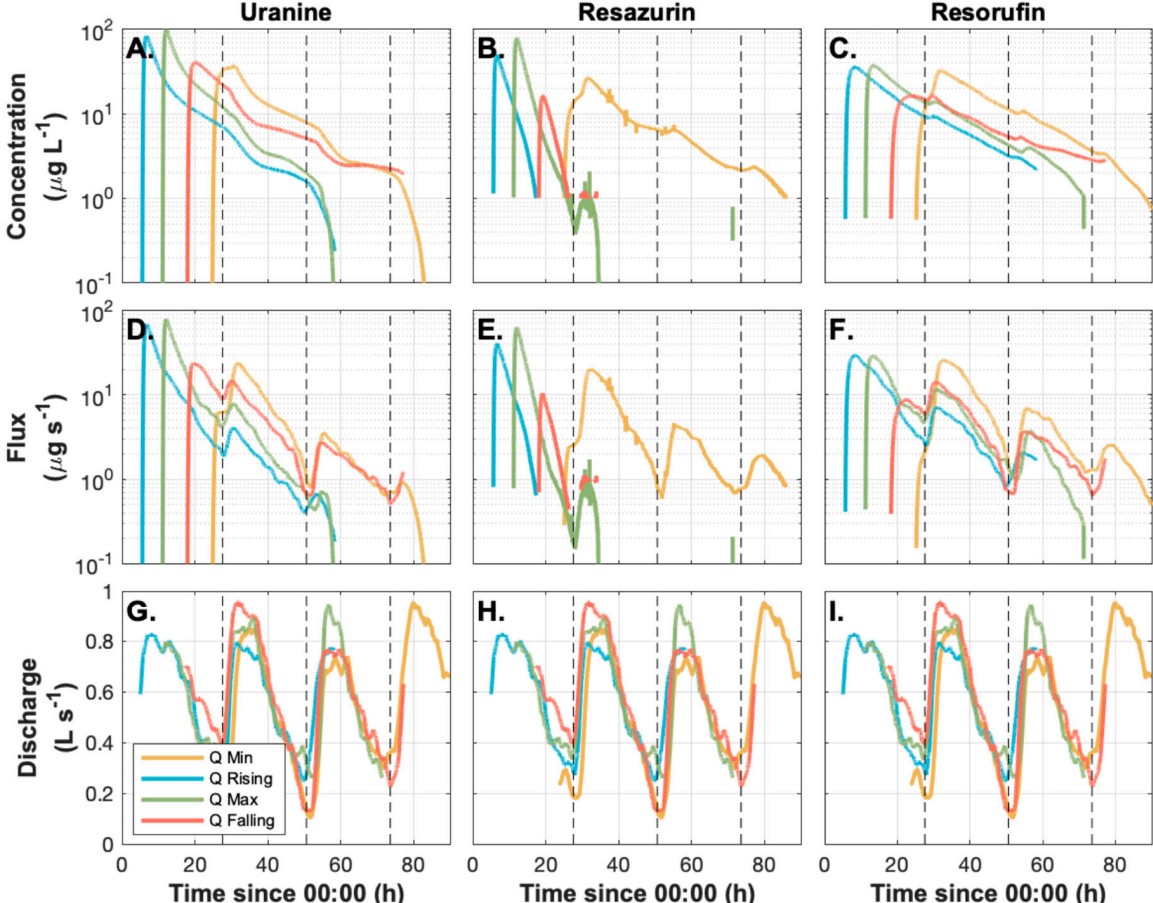

**Figure 2.** Time series of concentration (panels **A**–**C**), mass flux (panels **D**–**F**), and discharge (panels **G**–**I**) for all experiments, and all tracers including Uranine (conservative; panels A, D, E), Resazurin (parent compound; panel B, E, H), and Resorufin (daughter product; panels C, F, I). The dashed vertical lines indicate the time of lowest discharge.

Despite the differences in timing of advection of the highest concentration water relative to the time of minimum discharge, we found few differences between the injections with respect to calculated solute transport metrics. For example, we found no significant trends between discharge and any of the Ura breakthrough curve metrics (Table 2). Similarly, a single SAS function produced reasonable representations of the observed Ura time series for all injections ($\sigma_{error}$ = 3.2 µg L$^{-1}$). Moreover, all SAS model parameters were well constrained by the data. The $S_{min}$ was the parameter least sensitive to discharge. Both $Q_U$ and $S_U$ were well constrained and sensitive to discharge. The best-fit parameters vary by a factor of 4–6-fold (Table 3). However, the underflow time scale estimated as $S_U/Q_U$ varied from about 21 h at low flow to about 31 h at high flow. The SAS function varied systematically with discharge (Figure 3I–L and Figure 4). The long-term mean SAS function has an offset of about 2.5 m$^3$, representing the approximate plug flow volume in the study reach. The SAS function peaks around 15 m$^3$ of storage (the most frequently exported age-ranked storage), and then decays toward the oldest storage.

**Table 2.** Summary metrics for Uranine mass flux time series.

| Discharge at Time of Injection | $t_{peak}$ (h) | $t_{99}$ (h) | $M_1$ (h) | CV | Skewness | Apparent Dispersion ($\times 10^4$ m$^2$ h$^{-1}$) | Holdback |
|---|---|---|---|---|---|---|---|
| Falling | 2.3 | 55.3 | 14.3 | 0.85 | 1.35 | 1.55 | 0.61 |
| Rising | 1.7 | 45.0 | 8.5 | 1.04 | 1.80 | 1.39 | 0.68 |
| Peak | 1.4 | 39.3 | 8.2 | 1.01 | 1.48 | 1.26 | 0.68 |
| Minimum | 7.9 | 53.5 | 15.3 | 0.70 | 1.46 | 1.15 | 0.67 |

**Table 3.** Maximum likelihood parameter values from DREAM parameter estimation. Values in parentheses show the 10–90th percentile ranges. Columns show the value at median discharge, sensitivity (1/k), and parameter value at $Q_{10}$ (about 0.28 L s$^{-1}$) and $Q_{90}$ (about 0.83 L s$^{-1}$).

| | Parameter Values | | | Parameter Sensitivity |
|---|---|---|---|---|
| Parameter | Reference Value | Value at $Q_{10}$ | Value at $Q_{90}$ | $1/k_X$ * |
| Storage offset, $S_{min}$ (L) | 3336 (*3288, 3617*) | 3728 | 2981 | −1.25 (*−1.61, −1.33*) |
| Mean of gamma distribution, $\mu$ (L) | 29,063 (*28,981, 29,713*) | 37,406 | 22,513 | −1.10 (*−1.11, −1.02*) |
| Gamma distribution shape parameter, $\alpha$ (unitless) | 1.63 (*1.57, 1.64*) | 2.50 | 1.06 | −0.645 (*−0.717, −0.632*) |
| Underflow discharge, $Q_U$ (L s$^{-1}$) | 0.390 (*0.387, 0.405*) | 0.954 | 0.158 | −1.03E-3 (*−1.06E-3, −1.00E-3*) |
| Underflow scale parameter, $S_U$ (L) | 36,305 (*35,945, 38,837*) | 73,414 | 17,802 | −0.393 (*−0.403, −0.358*) |

* Values in each row are $1/k_X$, where $X$ is the parameter being described in each row. For example, the values in the top row are for $1/k_{S_{min}}$.

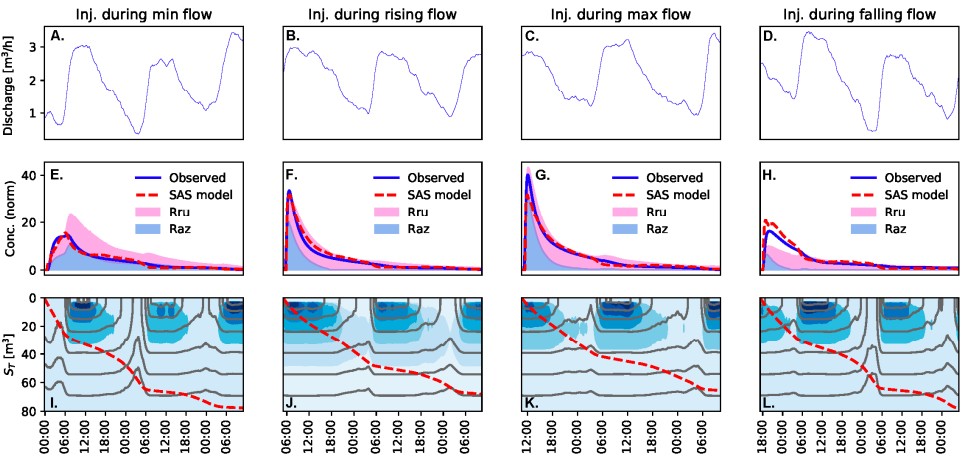

**Figure 3.** Time-series of discharge (panels **A–D**), observed and simulated concentration normalized by input mass (panels **E–H**), and density of the SAS function (panels **I–L**; darker color indicates higher probability). A vertical section along the bottom row can be interpreted as a probability density function of the age-ranked storage being exported from the study reach at any given time step.

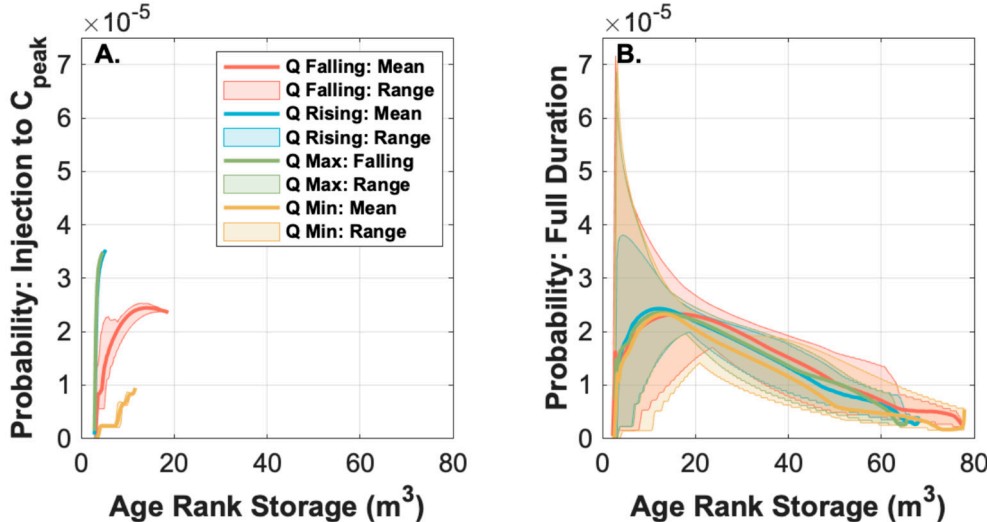

**Figure 4.** (**A**) Mean and range of SAS functions between tracer injection and the first diurnal peak in discharge after each injection. Note that the solute tracer injected at minimum discharge is too young to be exported prior to peak discharge, compared to all other injections which are biased toward export of the youngest age-ranked storage. (**B**) The full range of and overall mean SAS function for each injection. Time-variable SAS functions are displayed in the supplemental animation, showing how early-time disparities converge over the course of the entire time series.

During the rising, peak, and falling discharge injections, the tracer age was almost perfectly aligned with the ages being predominantly exported rather than stored. In contrast, tracer was too young to be exported rapidly during the minimum discharge injection. Thus, tracer injected at minimum discharge was predominantly stored in the reach, ultimately labeling older age-ranked storage that is more slowly and consistently exported from the study reach. These differences are attributable to the timing of the study release, as all four injections were well-described by a single SAS function and did not exhibit wide variation in summary metrics such as temporal moments. Indeed, the evolution of SAS functions, and particularly the conditions experienced shortly after each injection, may dominate overall transport and transformation dynamics in the study reach.

### 3.2. Observed Variation in Solute Transport is Primarily Due to Storage Release Timing Relative to an Oscillating Diurnal Cycle in the Underlying Transport Processes

The best-fit SAS function reveals that high-discharge periods are dominated by the release of younger water, and lower-discharge periods by the release of older water (Figure 3). Conceptually, this means periods of higher discharge preferentially shunt younger water through and out of the system, bypassing older storage. In contrast, during periods of low discharge, the water entering the upstream end of the study segment cannot be rapidly transported through and out of the reach. Instead, water entering during low-discharge conditions is almost entirely stored, and older water is displaced and released from the study reach. This means that the general principles of the pulse-shunt model [61] may be operating on a daily basis for a small headwater stream experiencing diurnal fluctuations in discharge. Overall, we found a single parameterization of the SAS function provided a strong fit to observed data, which indicates that the same consistent diurnal cycles of transport processes were present during all four injections and varied systematically with diurnal fluctuations in discharge. However, the observed in-stream time series demonstrate that the timing of the solute tracer release relative to the diurnal cycle is important in determining which age-ranked storage is sampled by the solute tracer, and how the tracer is ultimately stored in and released from the study reach.

We generally found that the occurrence of minimum discharge conditions serves as a diurnal reset on transport in the system. The observed Ura time series are systematically varied before reaching the first diurnal minimum discharge condition after injection (X = 24:00 in Figure 2). After this time of

minimum discharge, the slopes of the falling limbs are similar and responses of concentration and flux are synchronized with minimum discharge conditions. As such, we focus on the time between the injection and first minimum discharge occurrence as critical in defining differences between injections. Transport between the injection and minimum discharge condition, when youngest storage is mostly being exported from the system, are important determinants of the observed in-stream tracer time series in the first 24 h after 00:00 on the day of release.

### 3.3. Spatial and Temporal Variation in Raz-to-Rru Transformation

### 3.3.1. Rru Production Does Not Appear to Change with Discharge

How the coupled transport and transformation of Raz and the resultant Rru varies between injections is inconsistent with Ura. This inconsistency indicates that fluctuations in residence time are driving reaction progress rather than changes in reaction rates. For the rising, peak, and falling discharge conditions, in-stream Raz is predominantly transported through the reach prior to the first rising limb after injection (Figure 2B). In contrast, Raz injected during minimum discharge conditions is stored in the reach and slowly exported during the 2.5 days after its initial release. This is notable for two reasons. First, Raz is stored and released in substantially higher amounts than in other injections, with a time series more comparable to the conservative Ura tracer than the other Raz time series. Second, the late-time Raz is being exported in its unreacted form for the minimum discharge release, whereas all other injections have little-to-no measurable unreacted Raz after minimum discharge.

Reaction potential occurred throughout the study period, as evidenced by rapid Rru production in the study reach for all injections (Figure 2). The slopes of the rising limb of Rru production are similar, which we interpret as a comparable transformation rate across all injections occurring along the shortest and fastest flowpaths through the reach. Notably, peak Rru concentrations occurred with the rising limb for the injections at falling and minimum discharge (compared to pre-rising limb peaks for the rising and peak discharge injections). For the injection at minimum discharge, Rru continued to be exported from the study reach in approximately equal concentration to Raz for 60 h after the injection. In all other cases, Rru exceeded Raz (i.e., reacted tracer exceeds unreacted tracer) after the first rising limb of the hydrograph.

As a second line of evidence, ambient in-stream water chemistry exhibits no discernable trend with in-stream discharge (Figure 5; $p > 0.10$ for all solutes). We note here that in addition to discharge, other biogeochemically important factors likely varied on a diurnal basis, including dissolved oxygen, pH, solar radiation, and water temperature. We interpret the nearly steady-state concentrations for DOC, phosphate, and nitrate as indicative that no systematic shift in metabolism occurs in response to diurnal fluctuations in discharge or other factors (e.g., water temperature). Instead, we argue the long-term integration of the nitrogen, carbon, and phosphorous cycles in the system appears to be at approximately steady-state under the diel hydrologic forcing of flows in the stream. This interpretation is consistent with past studies in the same watershed, which concluded that biological limitations are the primary control on nutrient cycling during low flows [115].

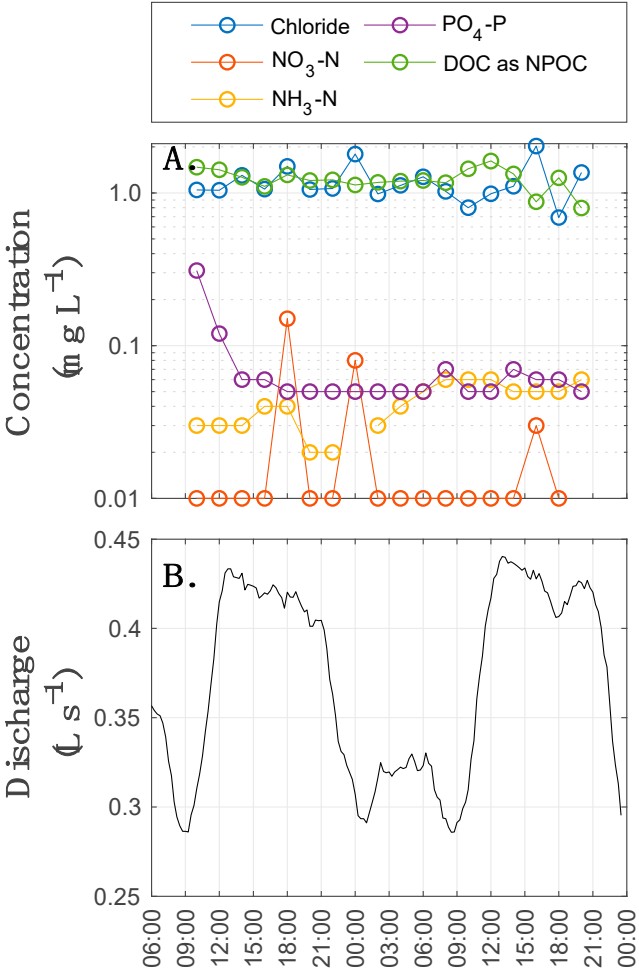

**Figure 5.** (**A**) Time series of nutrients and anions in the stream during diurnal fluctuations; (**B**) discharge during data collection.

### 3.3.2. Older Storage is Not as Metabolically Active as Younger Storage

While the transformation rates did not appear to be time-variable in our system, it does not mean that rates were spatially homogeneous. Instead, we found that both transport and transformation were controlled by injection timing, and ultimately the timescales of storage that were sampled by the tracers. During all injections, we observed rapid transformation of Raz-to-Rru associated with the youngest 20 h of age-ranked storage in the system (Figure 6A). However, during the three highest discharge injections, Raz was primarily transported through the study reach in less than 20 h, meaning that unreacted Raz largely bypassed storage within the reach rather than interacting with and documenting those domains. In contrast, the lowest discharge injection occurred at a time when water entering the study reach was stored rather than exported, allowing Raz to interact with and sample older age-ranked storage. Thus, the transformation of Raz-to-Rru is primarily associated with water spending less than 20 h in the study reach (Figure 6A), which corresponds to the youngest 40 m$^3$ of age-ranked storage in the reach (Figure 6B). Our observed higher transformation in the younger, presumably surficial, flowpaths can be explained by existing studies showing that the shallowest layer at the top of the streambed is the most metabolically active [75,116–118]. Further, this study demonstrates that the timing of tracer injection and concurrent flow conditions can influence the "window of detection" [118] of particular smart tracers for quantifying geochemical and ecological processes in the river corridor.

The persistence of unreacted Raz and the lack of Raz-to-Rru transformation in waters older than about 20 h (i.e., age-ranked storage 2–40 m$^3$) indicates that older storage is not metabolically active on the timescales relevant to the Raz additions (Figure 6). There are three possible explanations for

this behavior. First, it is possible that the redox conditions along these flowpaths do not favor the Raz-to-Rru conversion, which is primarily sensitive to aerobic and facultative anaerobic conditions [73]. However, we found bulk oxic conditions in the hyporheic water [114]. While anoxic microsites may exist, they are not likely to be abundant in the coarse streambed substrate of the site [119]. Overall, it is not expected that these bulk oxic conditions of the site would cause the expected inhibition of Raz transformation. Second, and more likely, the metabolic activity in the older age-ranked storage is low because of stoichiometric limitations of metabolic pathways affecting Raz. Similar to other detrital-based, heterotrophic streams, low nitrogen and phosphorus concentrations at our study site could limit metabolic activity [120]. The bioavailability of these nutrients likely limit microbial metabolism in our system under all flow conditions [116,121,122]. However, we would expect these limitations to be minimized along shallow, intermittently inundated, and hydrologically youngest flowpaths that experience warmer temperatures, are nearer to light exposure, increased reactive surface area, and availability of fresh organic matter sources. Lastly, metabolic activity may be energetically constrained on timescales relevant to the water residence time along these longer and hydrologically older flowpaths. Previous work has suggested that buried particulate organic matter from debris flows promotes hyporheic microbial respiration at the study site [122]. While this buried organic matter, likely western hemlock (*Tsuga heterophylla*) and Douglas fir (*Pseudotsuga menziesii*) wood and needles, may persistently increase metabolic activity of the oldest water at the site, it is also likely a low-quality organic matter source with less reaction potential, relative to the younger organic matter along the shallower hyporheic flowpaths, given its composition and old age within the stream corridor [123]. Together, there are multiple potential limitations on the Raz-to-Rru transformation that may exist in the older water at the site. Further work is needed to identify the location of the older hydrologic storage at the site, which will allow for the in situ sampling of the pore water chemistry needed to directly test these mechanistic constraints on Raz-to-Rru transformation.

While the exact location and distribution of the metabolically active storage within the study reach is unknown, we can ground our SAS-based interpretation with calculations of the known extent of hyporheic exchange in our study system. We estimate a surface water volume of 4.4 m$^3$ (typical width and depth of 0.8 and 0.1 m along the 55 m length) in the study reach, leaving approximately an additional 35.6 m$^3$ of metabolically active storage in the subsurface of the study reach (estimated as transformations in the youngest 40 m$^3$ of age-ranked storage minus the in-stream volume based on visual inspection of minimal transformation of Raz-to-Rru in older storage; Figure 6B). Notably, the SAS interpretation indicated about 2.5 m$^3$ as the plug flow volume, suggesting 1.9 m$^3$ of in-stream transient storage. On average, this would represent a metabolically active storage zone with a cross-sectional area of 0.65 m$^2$ along the study reach, which is about 8 times larger than the average cross-sectional area of the stream. This cross-sectional area is smaller than the broad penetration of tracer into the riparian zone observed during longer-duration injections monitored by electrical resistivity and in a network of monitoring wells [19–22]. However, this is consistent with a persistent, near-stream hyporheic zone [21] that would have been sampled by an instantaneous injection of tracer due to window of detection limitations [124,125]. Similarly, past studies found a metabolically active storage zone of about 2.45 times the area of the stream in a nearby headwater catchment [83].

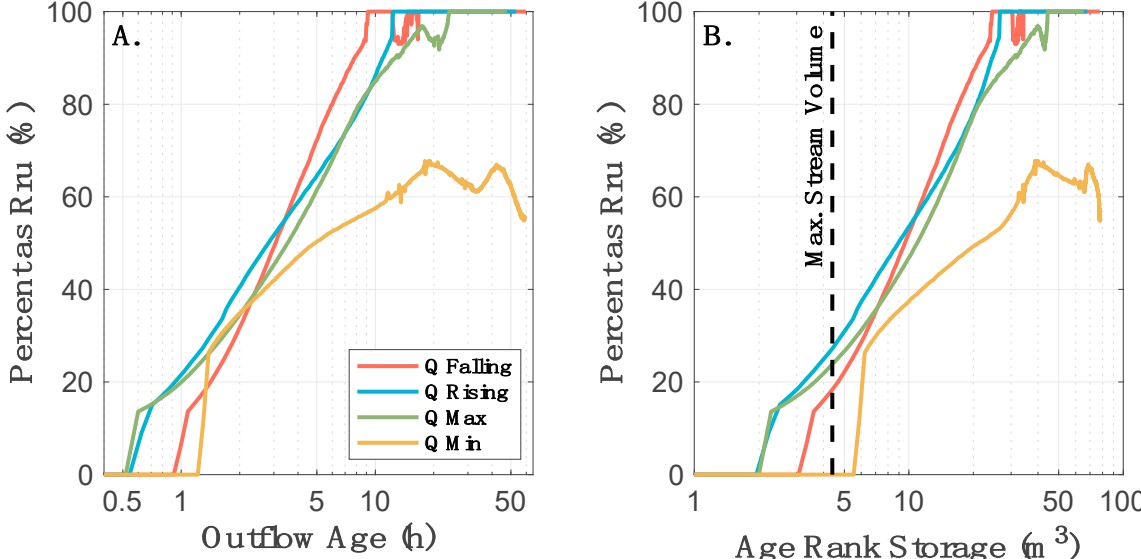

**Figure 6.** Fraction of outflowing Raz that has been transformed to Rru as a function of (**A**) outflow age, and (**B**) age-ranked storage in the study reach.

## 4. Conclusions

The objective of this study was to assess how reach-scale transport and transformation of reactive tracers vary in response to diurnal discharge fluctuations in an intermittent, headwater stream. Based on visual inspection, tracer transport varies depending upon discharge dynamics at the time of release into the system. However, a single time-variable SAS function was able to represent the variation with discharge in the system, and no summary metrics exhibited a significant trend with discharge. Thus, we conclude the transport patterns describe a single StorAge Selection function that varies with discharge, at least across the range of discharge observed in our study. Variation with time of release is visually apparent (Figure 2), but these differences are not sufficient to alter the overall storage and release of tracer at the timescales associated with the entire reach.

With respect to solute transformation potential of the stream, we expected diurnal fluctuations in discharge would alter contact times and overall observed Raz-to-Rru conversion in the system. Instead, we found little variation in Raz-to-Rru reaction rate through time. Our results suggest that the biological processes controlling the Raz-to-Rru transformation are static, and variations in the reactive transport of Raz are primarily due to variations in residence time and locations of water and solute storage, not transformation processes. This is in opposition to the simplest interpretation of the observed time series, which appear to indicate different transformation rates as a function of discharge (e.g., less transformation for the minimum discharge injection). However, with the aided insight of the SAS analysis, we see that, across all injections, Raz-to-Rru primarily occurs in the water with an outflow age of less than about 20 h, representing the youngest 40 m$^3$ of age-ranked storage in the reach. By injecting at different discharge conditions and exploring how age of stored water varied across those flow conditions, we were able to sample the metabolic activity of different storage locations within the system and show that the youngest storage has higher metabolic activity than the oldest storage. Therefore, not all storage in the stream has the same potential biogeochemical function, and increasing residence time of solute storage does not necessarily increase reaction potential of that solute.

**Supplementary Materials:** The following are available online at http://www.mdpi.com/2073-4441/11/11/2208/s1: Video S1: Evolution of SAS functions.

**Author Contributions:** Conceptualization, A.S.W., M.J.K., N.M.S., J.L.A.K., P.J.B., D.M.H., S.K., E.M., and J.P.Z.; data curation, A.S.W. and S.P.; formal analysis, M.J.K., N.M.S., J.L.A.K., C.J.H., J.D.D., M.M., and S.M.W.; funding acquisition, A.S.W., D.M.H., S.K., A.M., A.I.P., S.M.W., and J.P.Z.; investigation, M.J.K., N.M.S., J.L.A.K., P.J.B., J.D.D., A.L., K.N., M.M., S.P., and N.I.M.; methodology, M.J.K. and J.L.A.K.; project administration, A.S.W. and J.P.Z.; resources, A.S.W., D.M.H., S.K., E.M., A.I.P., and J.P.Z.; visualization, C.J.H.; writing—original draft, A.S.W.

and M.J.K.; writing—review and editing, M.J.K., N.M.S., J.L.A.K., P.J.B., C.J.H., J.D.D., D.M.H., S.K., A.L., E.M., A.M., K.N., M.M., S.P., A.I.P., N.I.W., S.M.W., and J.P.Z.

**Funding:** This research was funded by the Leverhulme Trust (Where rivers, groundwater and disciplines meet: A hyporheic research network), the UK Natural Environment Research Council (Large woody debris—A river restoration panacea for streambed nitrate attenuation? NERC NE/L003872/1), and the European Commission supported HiFreq: Smart high-frequency environmental sensor networks for quantifying nonlinear hydrological process dynamics across spatial scales (project ID 734317). Data and facilities were provided by the HJ Andrews Experimental Forest and Long Term Ecological Research program, administered cooperatively by the U.S. Department of Agriculture Forest Service Pacific Northwest Research Station, Oregon State University, and the Willamette National Forest and funded, in part, by the National Science Foundation under Grant No. DEB-1440409. Additional support to individual authors is acknowledged from National Science Foundation (NSF) awards EAR 1652293, EAR 1417603, EAR 1846855, and EAR 1446328, Department of Energy (DOE) award DE-SC0019377, and DOE's Office of Biological and Environmental Research via the Mercury Scientific Focus Area at Oak Ridge National Laboratory.

**Acknowledgments:** Any use of trade, firm, or product names is for descriptive purposes only and does not imply endorsement by the U.S. Government nor any author. Underlying data for this study are hosted by CUAHSI's HydroShare at: http://www.hydroshare.org/resource/d353ce4a7dc643c8a566504bfd578a57.

**Conflicts of Interest:** The authors declare no conflict of interest.

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
