# Peer review of "Solute Transport and Transformation in an Intermittent, Headwater Mountain Stream with Diurnal Discharge Fluctuations"

_water, doi:10.3390/w11112208_

Round 1

Reviewer 1 Report

It is a good work. However, the issues below should be clarified.

1.Diurnal hydrographys for rivers with infiltration have features similar to those associated with evapotranspiration, how you could attribute the observed diurnal discharge to evapotranspiration in the study reach?

2. Lack of explanations on riparian vegetation.

3. In addition to diurnal discharge variation, there is also diurnal water temperature variation that may affect solute transformation, especially during the low flow periods.

Besides, for discharge unit, better to use m3/s, instead of L/s

Reviewer 2 Report

GENERAL COMMENTS:

The paper presents a study of the role of time-varying discharge (the diurnal discharge variability) on the solute transport and transformations in the headwater mountain stream. Conservative (Uranine) and reactive (Resazurin, Resorufin) tracers were injected into the studied river section and the changes in the tracer concentrations were monitored. Generally, the topic covered in the manuscript suits well into the scope of the Water journal. In my view, the main drawback of the manuscript is related to the used methodology, especially the implementation of the StorAge Selection model which I found in some parts difficult to follow (details are provided below under specific comments). I believe readers not familiar with the StorAge model application will have difficulties in following the experimental results.

Additionally, in my view, one point is pretty much neglected in the manuscript. Daily discharge fluctuations considerably influence the hydrological connectivity at different spatial scales and daily variations of solutes concentrations. Namely, from the process point of view, during daily minimum flows, the longitudinal hydrological connectivity along the stream reach and between the stream channel and surrounding riparian areas could be often intermittent, whereas during the daily maximums, the hydrological connectivity along the stream channel becomes re-established. So how do these processes fit into the results presented in the manuscript, e.g. what could be the role of lost hydraulic connectivity at the injection points, which disables more rapid transport of the injected tracer in the time of daily minimum discharges. The authors mention the pulse-shunt concept (Raymond et al., 2016) but should further discuss the issue of the hydraulic connectivity the manuscript.

SPECIFIC COMMENTS:

Title:

I would suggest changing the title which is in my view awkward. In natural streams the discharge is temporally highly unsteady variable, even under hydrologically stable (rainless) conditions as the ones considered in the manuscript. Therefore, I would suggest changing the title: “Solute transport and transformation in an intermittent, headwater mountain stream.”

Abstract:

Line 41: Change “locatiosn” to “locations”.

The authors should highlight that the study was conducted during hydrologically relatively stable (e.g. rainless) conditions in the abstract.

Keywords: I would suggest changing “unsteady-state discharge” to e.g. “discharge variability or discharge dynamics”.

Introduction:

Introduction section is informative and relatively well written but it could be more concise in some parts.

Lines 62-71: What about the influence of hydrologically induced fluxes from riparian areas and from wider surrounding catchment areas? The daily discharge variability is only in a minor extent controlled by the in-stream processes.

Line 105-106: Aren’t the conditions the authors conveyed their research low-flow conditions but with evident diurnal discharge pattern? What do the authors mean by “steady” conditions?

Methods:

Lines 178-180: At what time did the daily minimum and maximum flows occur?

Equations: Please double check that all parameters used in equations are properly explained in the accompanying text. E.g.:

Line 231 (equation 6): What is “F”?

Line 232 (equation 7): What is “Tav”?

Line 235 (equation 8): What is “L”?

Line 257: What is underflow discharge Qu? I believe people not familiar with the StorAge Selection will have difficulties in trying to understand the interpretation of results.

Results and Discussion

Line 307: “High flows” sounds like discharge increase after rainfall events. I would suggest using “daily maximum flow”.

Table 2: Please explain variables Tpeak and T99 in table’s caption.

Line 329, line 330: Do you have any data which could support your statement about the plug-flow volume and the exported age-ranked storage?

Table 3: I found the DREAM parameter values estimation data presented in table of little information value. The results should be further referred and in the text.

Figure 4: Could authors show some probability bounds (e.g. 90 or 95%)?

Line 356: I generally agree with the point the authors are trying to discuss. However, how would authors argue statement that the variations in the solute transport due to storage release timing are controlled by the underlying transportation processes? Which underlying transportation processes are meant here?

Lines 404-406: Do the authors have any other data (e.g. the dissolved oxygen and pH measurements), do they have and observable daily pattern (which is often the case)?

Line 455: Comma is doubled.

Lines 464-467: How was the subsurface storage volume estimated (e.g. the youngest 40 m3 of age-ranked storage)? What is the uncertainty of such estimation?

Line 469: Cross section area of the entire stream channel or up to a certain discharge conditions?

Round 2

Reviewer 1 Report

The cited references [25, 104, 105] were published in 2002, 2008 and 2007, respectively. They are kind of old. Are there any hard evidences proving the diurnal discharge fluctuations are still driven by evapotranspiration today?  In the cited work [25], the measurement of vegetation cover was conducted in 1999. How you can be so sure that the riparian vegetation community is not essential to this study because the vegetation conditions in the 1999s are available in literature.   

Is the diurnal water temperature variation irrelevant?

Round 3

Reviewer 1 Report

Added explanations were accepted